# Electrospray Deposition of Cellulose Nanofibers on Paper: Overcoming the Limitations of Conventional Coating

**DOI:** 10.3390/nano12010079

**Published:** 2021-12-29

**Authors:** Quim Tarrés, Roberto Aguado, M. Àngels Pèlach, Pere Mutjé, Marc Delgado-Aguilar

**Affiliations:** LEPAMAP-PRODIS Research Group, University of Girona, M Aurèlia Capmany, n°61, 17003 Girona, Spain; joaquimagusti.tarres@udg.edu (Q.T.); angels.pelach@udg.edu (M.À.P.); pere.mutje@udg.edu (P.M.); m.delgado@udg.edu (M.D.-A.)

**Keywords:** barrier properties, cellulose nanofibers, electrospray deposition, nanocellulose, papermaking, strength agent

## Abstract

While the potential of cellulose nanofibers to enhance the mechanical and barrier properties of paper is well-known, there are many uncertainties with respect to how to apply them. In this study, we use not only bulk addition of micro-/nanofibers and bar coating with oxidized nanofibers, but also a combination of these and, as a novel element, electrospray deposition of nanofiber dispersions. Characterization involved testing the strength of uncoated and coated paper sheets, their resistance to air flow, their Bendtsen roughness, and their apparent density, plus visualization of their surface and cross-sections by scanning electron microscopy. As expected, bulk addition to the unrefined pulp was sufficient to attain substantial strengthening, but this enhancement was limited to approximately 124%. Following this, surface addition by bar coating improved air resistance, but not strength, since, as applying nanocellulose at high consistency was technically unfeasible, this was performed several times with detrimental drying stages in between. However, replacing bar coating with electrospraying helped us overcome these apparent limitations, producing enhancements in both barrier and tensile properties. It is concluded that electrosprayed nanofibers, owing to their uniform deposition and favorable interactions, operate as an effective binder between fibers (and/or fines).

## 1. Introduction

Nanocellulose, especially in the forms of cellulose nanofibers (CNFs) and cellulose micro-/nanofibers (CMNFs), has been generating increasing interest among researchers and manufacturers [1,2,3]. From the academic perspective, despite numerous articles (mainly published from 2010 to today) that have involved CMNFs in some form [4,5], the field is still full of uncertainties with respect to potential applications. The possibilities seem endless, from reinforcement of structural materials [6], to providing all kinds of electronic devices, such as transistors and nanogenerators [7,8], with flexibility and transparency. From the industrial perspective, the nanocellulose market is expected to increase at an annual growth rate of up to 21.3% [9]. Unsurprisingly, among all the industries that could potentially benefit from the use of this renewable nanomaterial, paper companies play an important role [10]. Indeed, cellulose micro-and/or nanofibers are known to be outstanding strengthening agents when used as a bulk additive [11,12] prior to the formation of the paper web. If, besides or instead of strengthening, the primary goal is to confer effective barrier properties, the surface addition of nanocellulose (as a coating component) seems to be a promising approach [13,14].

Traditionally, paper has been deemed a low value-added commodity. Such consideration still holds true, given that, at least among European papermakers, value added amounted to only 22% of the total turnover in 2019 and 2020, and it has been like this since 2010 [15]. However, in our judgment, CMNFs offer an unmissable chance to increase market advantages, particularly for packaging paper. They do so, first, by providing customers and delivery companies with a material whose strength matches that of composites consisting of paper and non-biodegradable resins [16,17]. Second, they do so by decreasing porosity and permeability, and thus enhancing the barrier properties of the material towards air, grease, water vapor, and even liquid water [11,17,18,19]. In short, nanocellulose-reinforced paper has the potential not only to partly replace conventional packaging paper, at least for special deliveries, but also to progress the substitution of non-biodegradable, single-use plastics [20,21].

Nonetheless, CNF-reinforced paper has severe limitations as a composite material, mainly arising from the fact that both the matrix and the reinforcing phase of the material are cellulosic. On the one hand, it is difficult to imagine that the water resistance of a fully cellulosic material could match those of a polymer resin, such as polyethylene terephthalate. On the other hand, when it comes to its mechanical properties, certain improvements have been made since Eriksen et al. obtained increments of up to 30% in the tensile strength of thermomechanical pulp handsheets (for which relative increments are generally lower than for chemical pulps) [22]. One of them is taking advantage of the synergy between cationic starch (CS) and nanocellulose, as highlighted in a recent review [14], and thus using CS as a retention agent. Other strategies are related to the chemical/enzymatic/mechanical treatments that usually precede nanofibrillation [23]. For instance, (2,2,6,6-tetramethylpiperidin-1-yl)oxyl (TEMPO)-mediated oxidation enables ease of dispersion in water, but at the cost of slowing down dewatering to such an extent that this pretreatment is deemed unfeasible for bulk addition [24]. As advocated in a previous work, TEMPO-oxidized CNFs should be confined to coating, whereas glucanase-hydrolyzed CMNFs achieve promising results when combined with the pulp stock before sheet formation [25,26].

In this study, we suggest a combined approach, encompassing simultaneous bulk addition of CMNFs and surface addition of CNFs, to enhance paper strength and air resistance. As a novel approach, we consider the hypothesis that surface addition by electrospraying, instead of conventional coating, can overcome the limitations of the typical procedures used for nanocellulose-reinforced paper. The basis of this hypothesis is that momentum transfer is severely hindered in dispersions of nanoscale oxycellulose, but the application of an external electric field allows control of the flowrate of droplets over a surface. Electrospray deposition has seldom been used on paper, although some applications of poly(lactic acid) nanoparticles on paperboard have been considered [27,28]. Partially because oxidized CNFs have lengths in the nanoscale without the need of electrical stimuli, and partially because electrospray deposition still seems unthinkable in a paper mill, this technique has been disregarded, if ever considered, for paper coating with nanocellulose. Partly inspired by the success of bioplastic nanomaterials, we compared the effects of electrospray deposition of CNFs on paper to the effects attained by bar coating, and both coating techniques to bulk addition alone. Characterization of reinforced sheets involved mechanical tests, Gurley porosity essays, scanning electron microscopy, Bendtsen roughness measurements, and routine determinations of the apparent density and the basis weight.

## 2. Materials and Methods

### 2.1. Materials

Production of both types of cellulose nanofibers, TEMPO-oxidized nanofibers, glucanase-hydrolyzed nanofibers, and paper sheets were carried out using an unrefined bleached hardwood kraft pulp (BHKP) supplied by Torraspapel S.A. (Sant Joan les Fonts, Girona, Spain) as raw material. The bleached eucalyptus fibers used originally had an average length of 700 µm and an average diameter of 16 µm, as measured by means of a MorFi Compact analyzer (TechPap, Gières, Isère, France).

The chemical reagents used for TEMPO catalyzed oxidation were sodium bromide 99%, sodium hypochlorite 15%, sodium hydroxide 0.5 M, and TEMPO catalyst. All reagents were purchased from Merck KGaA (Darmstadt, Germany) and used without further purification.

Production of the glucanase-hydrolyzed nanofibers was carried out by a 2% endo-β-1,4-glucanases commercial enzyme cocktail Novozym 476 (Novozymes A/S, Copenhagen, Denmark), with an activity factor of 4500 CNF-CA/g cellulose (obtained on a sodium carboxymethyl cellulose substrate).

The dual retention system used in the nanofibers bulk addition was composed of cationic corn starch, C*Bond HR, supplied by Cargill Ltd. (Conover, IA, USA), with a degree of substitution of 0.042–0.049, and colloidal silica, with particle size in the range 8–30 nm and a density of 1.2 g/cm^3^, provided by the paper company LC Paper S.A. (Besalú, Spain).

A general scheme of the experimental procedure is presented in Figure 1.

### 2.2. Cellulose Nanofibers Production

The pretreatment of BHKP for the glucanase-hydrolyzed nanofibers production was conducted in an atmospheric reactor at 50 °C, following the procedure described by Tarrés et al. [29]. Briefly, fibers were suspended in water at 5 wt.% pulp consistency and the pH was adjusted to 5 by the addition of 0.1 N hydrochloric acid. The enzyme cocktail was added to the pulp at 50 °C and constant agitation at a dosage ratio of 300 g/tonne. The pretreatment time was 2 h and then the enzymes were deactivated by heating the suspension at 80 °C for 15 min and subsequent washing.

Bleached eucalyptus fibers were oxidized by the addition of 15 mmol of sodium hypochlorite per gram of cellulose. The TEMPO-catalyzed oxidation was performed in the presence of 0.1 g NaBr and 0.016 g TEMPO per g of fibers, according to the methodology described by Saito et al. 2007 [30]. The oxidation was performed at constant pH 10 by the addition of sodium hydroxide after the addition of sodium hypochlorite. Finally, the oxidized fibers were washed and filtered repeatedly to neutral pH.

Both pretreated fibers were homogenized in PANDA Plus high-pressure homogenizer (Gea Niro Soavi, Parma, Italy). The homogenization process was carried out by passing a 1 wt.% suspension of pretreated fibers by a sequence of 3 passes at 300 bar, 3 passes at 600 bar, and 3 passes at 900 bar.

The characterization of the nanofibers obtained was presented in previous works [25,31]. The endoglucanase-hydrolyzed nanofibers showed a carboxyl rate of 42 µeq/g, a cationic demand of 263 µeq/g, and a nanofibrillation yield of 34.0%. The TEMPO-nanofibers presented 1278 µeq/g, 1850 µeq/g and 98.89%, respectively.

### 2.3. Cellulose Nanofibers Addition and Paper Properties Characterization

Endoglucanase-hydrolyzed micro-/nanofibers were added in bulk at 0, 1.5, 3, and 4.5 wt.%, based on dry pulp weight. Then, a dual retention system (0.8 wt.% of cationic starch and 0.5 wt.% of silica) was added maintaining the pulp suspension, whose consistency was 0.2 wt.%, under constant stirring. Finally, paper sheets of approximately 76 g/m^2^ were produced in a Rapid Köthen paper sheet former (I.S.P., S.L., Gipuzkoa, Spain).

The surface application of TEMPO-oxidized nanofibers was conducted by two different systems, bar coater (RK Printcoat Instruments, Royston, UK) and electrospraying (Bioinicia, Valencia, Spain). In both cases, applications were carried out to obtain three degrees of coating, based on the increase in basis weight (1–3 g/m^2^). In the case of the bar coater, a suspension of nanofibers at 0.45 wt.% was used in one, two, or three layers. The sheet was dried after each application. The electrospray application was carried out using a suspension of nanofibers at 0.3% for 20, 40, or 60 min.

Air drying was carried out by means of a thermo-ventilator (50–57 °C, 30 min), while partially restraining shrinking by attaching the sheets to a steel surface. Then, sheets, still with a moisture content above 20 wt.%, were briefly hot-pressed at 93 °C in the drier of the aforementioned Rapid Köthen system (1 min).

The paper structure and the coating characteristics were analyzed by scanning electron microscopy (SEM) (ZEISS DSM 960A, Madrid, Spain). The evolution of air resistance was determined by Gurley porosity test following the methodology described in the standard [32]. Tensile properties of the uncoated and coated papers, such as breaking length, Young’s modulus and elongation, were measured by a universal testing machine (Instron, Barcelona, Spain) under standard conditions [33,34]. Roughness of paper sheets was analyzed by means of a Bendtsen tester (IDM Test, San Sebastian, Spain), following the ISO standard 8791-2 [35].

## 3. Results

### 3.1. Bulk Addition of Cellulose Micro- and Nanofibers

Micrographs of paper sheets containing up to 4.5 wt.% CMNFs are displayed in Figure 2. Micro-/nanofibers, at magnifications of 150× (cross-sectional views) or 1000× (surface views), displayed in micrographs a distinguishable clear tone, i.e., they reflected more electrons. From Figure 2d–f, the increasing amount of CMNFs across the paper sheet is clear. Similarly, bulk addition favored a compact structure, binding fibers together along the Z direction.

The effect of bulk addition on important mechanical and surface properties is presented in Table 1. While total retention of CMNFs in the sheet cannot be assumed, since CMNFs are too small to be mechanically retained [10,36], the insignificant differences in basis weight allow the conclusion that the retention aid system succeeded in retaining most of the CMNFs. Regarding the surface, even without coating, Bendtsen roughness was slightly but consistently increased by the bulk incorporation of CMNFs.

Figure 3 shows the enhancement of breaking length and air resistance (i.e., the decrease in Gurley porosity). As predicted, breaking length was correlated with other tensile properties that are reported in Table 1.

### 3.2. Surface Treatments with Oxidized Cellulose Nanofibers

In order to envisage the way oxidized CNF-based coatings interact with fibers and fines on the surface of paper sheets, Figure 4 presents 12 micrographs at different levels of magnification. Six of them (Figure 4a–f) correspond to the surface of sheets with CNF layers coated by different means or with no coating at all. While the surface binding effect after CNF-based bar coating can be easily appreciated in Figure 4c, electrospray deposition provided a progressive homogenization of the surface, as can be seen from Figure 4d–f. In the latter case, the whole surface is covered by the CNF dispersion.

Cross-sectional SEM images of sheets are displayed in Figure 4g–i. The Z-direction compacting effect of bulk addition (Figure 4g) was not present in sheets with oxidized CNFs applied only on the surface, be it by bar coating (Figure 4h) or by electrospraying (Figure 4i). Nonetheless, a much higher level of magnification (9000×) allows for cross-sectional views of individual fibers (Figure 4j–l). These micrographs were taken near the surface of the sheets. Only in the case of electrospray deposition were superficial fibers surrounded by CNFs (Figure 4l). Additional micrographs are provided in Appendix A.

#### 3.2.1. Limitations of Bar Coating in the Enhancement of Paper Properties

The high viscosity of oxidized CNF dispersions imposes a restriction on the quantity of nanofibers that can be placed on the surface of paper in a single application, as high consistencies carry evident problems of runnability. This is why, as described above, bar coating was performed as multiple layers, but this multistage treatment implied drying after every application. The rate of this drying, due to the hydrophilic character of nanofibers, was slowed down with increasing CNF content [37]. Table 2 and Figure 5 present the influence of CNF-based bar coating on paper strength, air permeability, roughness, and density. There was a consistently positive trend in air resistance (i.e., a decrease in surface porosity), especially when combined with bulk addition of CMNFs. However, effects on tensile properties were somewhat unpredictable.

#### 3.2.2. Electrospray Deposition

While conventional coating processes imply the use of excess suspension and doctoring away or recycling the excess, electrospray deposition allows for nearly total material efficiency; almost all the small drops generated by liquid atomization end up, slowly, covering the surface of paper. The nearly sealed surface that can be recognized in Figure 4f matched the increase in both apparent density (Table 3) and air resistance (Figure 6a) with an increase in elapsed time of electrospray deposition. The Gurley air resistance reached values over 300 s, which is still in the medium range but significantly higher than the unrefined reference sheets. The consistently positive effects on key mechanical properties, namely Young’s modulus, elongation (Table 3) and breaking length (Figure 6), are also worth mentioning, at least relative to the CMNF-free, unrefined, control pulp. While the reduction in surface porosity, at the cost only of a slight increase in roughness, was expected, the strengthening capabilities of this alternative coating method, attaining breaking length values that tripled the performance of the unrefined reference pulp, stands as a key finding of this work.

## 4. Discussion

It has been shown, by micrographs and by the positive effects on air resistance, that bulk addition of CMNFs had a compacting effect on paper sheets, particularly along the Z direction. Likewise, strengthening was obvious, especially from 0% to 3%, by which the breaking length roughly doubled its value (up to 3.6 km). These results support the positive effects on paper strength that have previously been reported in the literature [38]. It should be noted that this enhancement of paper strength is in the range of what could be expected from refining to more than 1500 PFI revolutions [39]. Both refining and CMNF addition increase the specific surface area and thus non-covalent (hydrogen) bonding interactions, the main factor influencing paper strength.

Bar coating consistently helped decrease porosity at the surface. Thus, the surface application of anionic nanofibers protected the sheet from air flow for longer in the air permeability (Gurley) test, up to 197 s. However, effects on tensile properties were not unambiguous. Even combining bulk incorporation and coating, breaking length did not go beyond 4 km. In fact, bar coating of the most resistant reinforced sheets had a negative impact on the performance in tensile tests. This loss can be partially explained by wetting and re-drying up to three times. During drying, fibers approach each other, closing or tightening the pores and decreasing the specific surface area [40]. The fiber wall lamellae tend to collapse too, leading to shrinkage of the fiber and, when considering both intra- and inter-fiber effects, of the paper sheet itself, in concordance with the higher roughness observed. Moreover, the more nanofibers are in/on the paper, the more water is required to be evaporated and the more severe shrinkage is.

With respect to the original contribution of this study, electrospray deposition allowed oxycellulose nanofibers to bind in an effective way to the superficial fibers and fines of paper resulting in the apparent homogeneity of the surface in micrographs (Figure 4d–f). In consequence, air permeability was decreased significantly, indicated by a significantly higher Gurley air resistance value compared to the reference case (over 300 Gurley s vs. 1.5 Gurley s). The resulting values were in the same range as those of Mousavi et al. [41] for rod-coating of refiner CNFs with a similar weight gain, but one order of magnitude lower than when they added carboxymethyl cellulose and/or obtained higher coat weights.

The most notable achievement was not the enhancement of air barrier properties, but increase in paper strength (in relative terms). This is not to imply that a strengthening coating layer is an unexpected outcome, since positive effects on mechanical properties have been observed even for conventional coating formulations [42]. In these cases, the coating suspension contained components different from those of the paper substrate itself. Considering that the uncoated sheet already contained CMNFs, and also comparing this enhancement with that achieved by bar coating, it may be concluded that electrospray deposition allowed for a distribution of CNFs that cannot be attained by conventional sheet forming and coating processes. Whereas multistage bar coating was subjected to the negative consequences of re-drying (shrinkage), electrospraying placed CNFs ubiquitously across the surface and even beneath it, judging by the increase in apparent density. This property saw an increment of +0.08 g cm^−3^ from uncoated papers with 60 min of electrospray deposition. This is consistent with the micrograph at Figure 4l, in which the fiber is surrounded on both sides by the dispersion of nanofibers.

## 5. Conclusions

To the question of which stage of papermaking offers the best opportunity to enhance the properties of the end-product, either bulk addition of surface treatment, this study provides a clear answer—both. However, bar coating, a rather conventional surface treatment process, only succeeded in sealing the paper against air flow, while there were limited or even adverse effects on paper strength. The most plausible reason lies in the need for re-drying after every coating layer, given the limitations of a single application. Electrospraying was able to deposit at least as much of the CNF dispersion as 3 layers of bar coating with only one post-drying step. The applied nanofiber layer not only slowed down air flow, attaining a Gurley air resistance over 300 s, but also led to a significant increase in the breaking length, especially when compared to the unrefined reference handsheets with no bulk addition of CMNFs. Overall, atomization by electrical stimuli overcame the large limitations to momentum transfer that CNF suspensions usually imply in a single application case and avoided fiber collapse by re-drying in the case of multilayer coating. While electrospray deposition is currently not feasible for a paper machine, an adaptation of the concept to the existing technology (e.g., spray nozzles) is worth pursuing.

## Figures and Tables

**Figure 1 nanomaterials-12-00079-f001:**
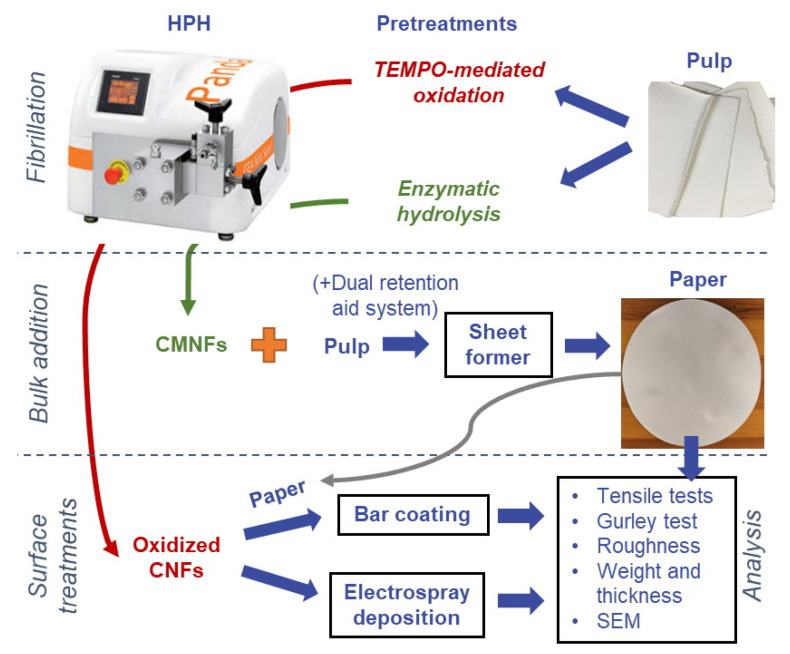
Diagram of the general experimental procedure, including fibrillation, sheet forming, bar coating, and electrospray deposition.

**Figure 2 nanomaterials-12-00079-f002:**
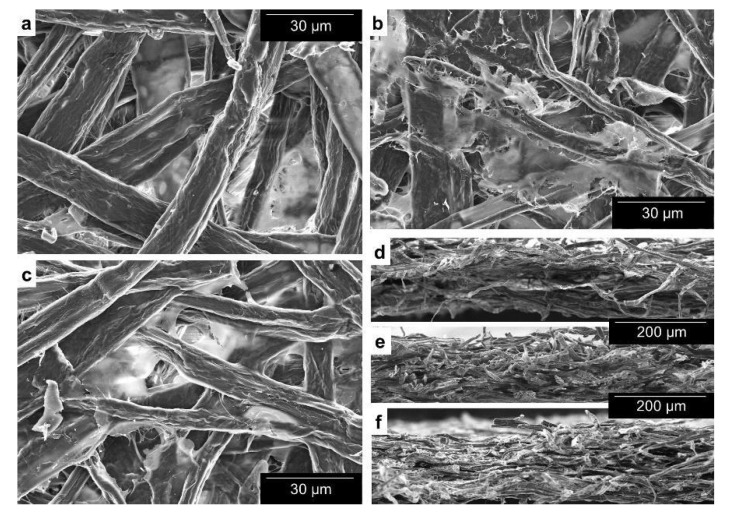
Visualization of the effect of bulk addition. SEM images of: (**a**) the surface of paper, for a magnification of 1000×, without any micro-/nanofibers addition; (**b**) the surface of paper with 1.5 wt.% CMNFs (in bulk); (**c**) the surface of paper with wt.4.5% CMNFs (in bulk); (**d**–**f**) cross-sectional views corresponding to (**a**–**c**), respectively, at a magnification of 150×.

**Figure 3 nanomaterials-12-00079-f003:**
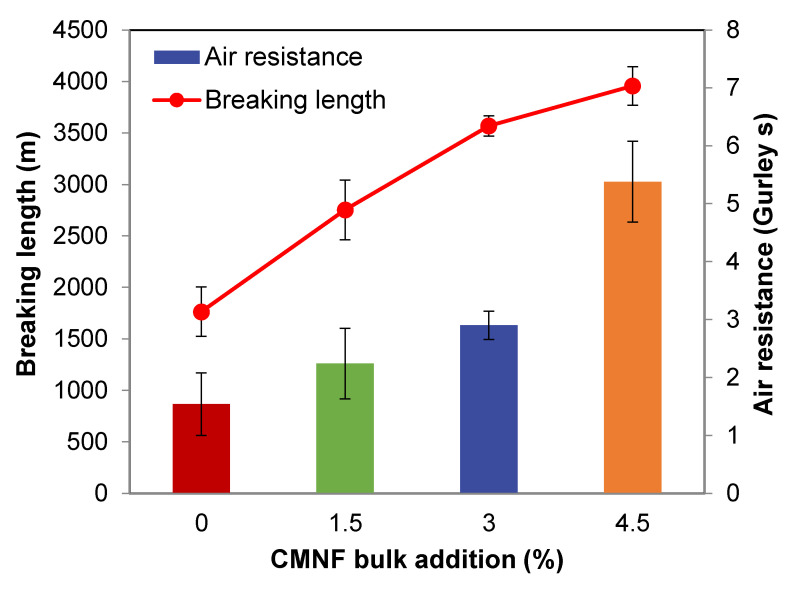
Evolution of breaking length (line) and air resistance (columns) with the proportion of micro-/nanofibers in bulk. Error bars correspond to twice the standard deviation.

**Figure 4 nanomaterials-12-00079-f004:**
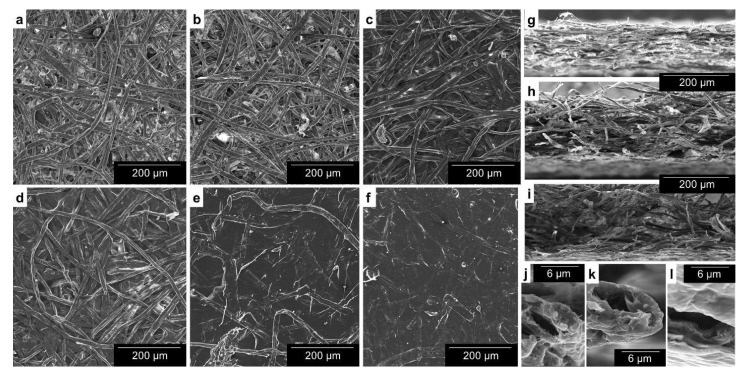
Visualization of the effect of surface treatments. SEM images of: (**a**) the surface of paper, at a magnification of 150×; (**b**) the surface of paper with 3 wt.% CMNFs (in-bulk) for comparison purposes; (**c**) the surface of paper without CMNFs in bulk, but bar-coated with three layers of CNFs; (**d**–**f**) the surface of paper sheets without CMNFs in bulk but coated by electrospray deposition of CNFs during 20, 40 and 60 min, respectively, at a magnification of 150×; (**g**–**i**) cross-sectional views of (**a**,**c**,**d**) sheets, respectively, at a magnification of 150×; (**j**–**l**) cross-sectional views of fibers corresponding to control paper, bar coating and electrospray deposition, respectively, at a magnification of 9000×.

**Figure 5 nanomaterials-12-00079-f005:**
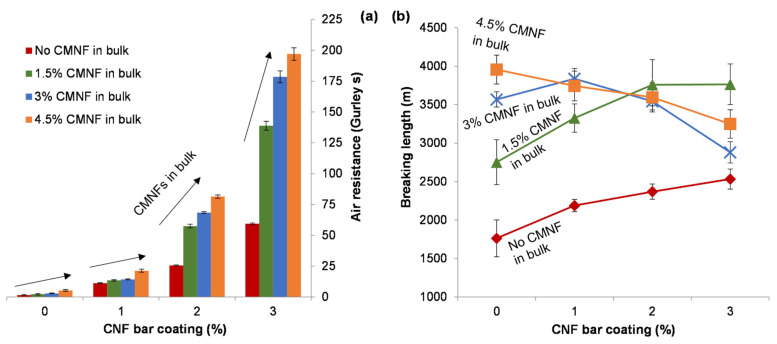
Evolution of air resistance (**a**) and breaking length (**b**) with the concentration of oxycellulose nanofibers in the coating suspension (bar coating), for the four levels of enzymatic CMNF content in bulk. Error bars correspond to twice the standard deviation.

**Figure 6 nanomaterials-12-00079-f006:**
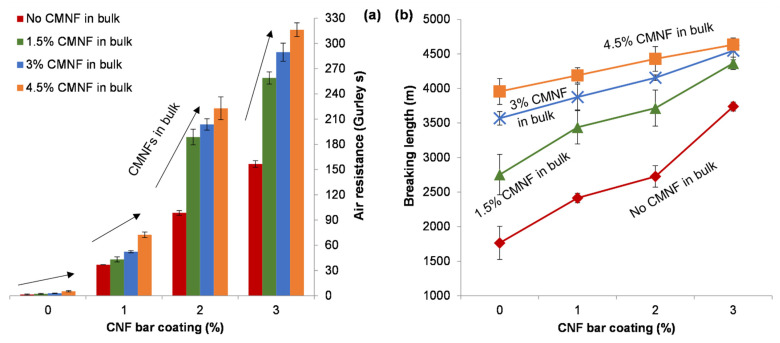
Evolution of air resistance (**a**) and breaking length (**b**) with time of electrospray deposition from a dispersion of oxycellulose nanofibers, for the four levels of CMNF in-bulk content. Error bars correspond to twice the standard deviation.

**Table 1 nanomaterials-12-00079-t001:** Paper properties, other than those represented in Figure 3, of sheets containing micro-/nanofibers in bulk. The amplitude of the intervals is twice the standard deviation.

CMNF Bulk Addition (%)	Basis Weight (g/m^2^)	Roughness (NmL/min)	Apparent Density (g/cm^3^)	Young’s Modulus (GPa)	Elongation (%)
0	77.0 ± 0.6	2160 ± 29	0.54	2.50 ± 0.27	0.61 ± 0.04
1.5	75.8 ± 0.5	2210 ± 7	0.57	2.96 ± 0.18	1.26 ± 0.08
3	75.6 ± 0.6	2260 ± 20	0.58	3.46 ± 0.23	1.43 ± 0.16
4.5	78.1 ± 0.6	2325 ± 12	0.59	3.62 ± 0.05	1.89 ± 0.10

**Table 2 nanomaterials-12-00079-t002:** Paper properties, other than those displayed in Figure 5, of sheets containing both micro-/nanofibers in bulk and nanofibers applied by bar coating. The amplitude of the intervals is twice the standard deviation.

CMNF Bulk Addition	CNFs Layers	Roughness (NmL/min)	Apparent Density (g/cm^3^)	Young Modulus (GPa)	Elongation (%)
0	1	2260 ± 16	0.59	3.15 ± 0.05	0.80 ± 0.07
	2	2310 ± 16	0.61	3.29 ± 0.25	0.80 ± 0.08
	3	2240 ± 10	0.61	3.06 ± 0.07	1.84 ± 0.11
1.5	1	2330 ± 7	0.61	3.34 ± 0.04	1.90 ± 0.05
	2	2296 ± 5	0.62	3.56 ± 0.17	1.88 ± 0.05
	3	2356 ± 5	0.62	3.27 ± 0.23	1.37 ± 0.04
3	1	2296 ± 5	0.62	3.56 ± 0.17	1.88 ± 0.05
	2	2356 ± 5	0.63	3.27 ± 0.23	1.37 ± 0.04
	3	2447 ± 4	0.64	1.83 ± 0.10	1.71 ± 0.08
4.5	1	2374 ± 8	0.63	3.31 ± 0.13	1.98 ± 0.10
	2	2438 ± 12	0.65	3.30 ± 0.12	1.64 ± 0.10
	3	2478 ± 6	0.65	2.86 ± 0.11	1.41 ± 0.08

**Table 3 nanomaterials-12-00079-t003:** Paper properties, other than the ones shown in Figure 6, of sheets containing both micro-/nanofibers in bulk and a layer of nanofibers applied by electrospray deposition. The amplitude of the intervals is twice the standard deviation.

CMNF Bulk Addition	Electrospray Time (min)	Roughness (NmL/min)	Apparent Density (g/cm^3^)	Young’s Modulus (GPa)	Elongation (%)
0	20	2219 ± 23	0.58	2.65 ± 0.10	0.84 ± 0.10
	40	2269 ± 16	0.60	2.91 ± 0.04	1.18 ± 0.12
	60	2314 ± 19	0.66	3.83 ± 0.10	1.33 ± 0.03
1.5	20	2249 ± 20	0.59	3.21 ± 0.10	1.42 ± 0.05
	40	2309 ± 27	0.62	3.65 ± 0.21	1.57 ± 0.10
	60	2360 ± 9	0.67	3.90 ± 0.21	1.84 ± 0.05
3	20	2319 ± 10	0.60	3.81 ± 0.02	1.53 ± 0.09
	40	2354 ± 25	0.63	4.10 ± 0.13	1.62 ± 0.08
	60	2460 ± 23	0.68	4.18 ± 0.07	1.97 ± 0.02
4.5	20	2379 ± 24	0.61	4.19 ± 0.07	1.95 ± 0.06
	40	2454 ± 27	0.64	4.48 ± 0.12	2.06 ± 0.09
	60	2490 ± 12	0.69	4.57 ± 0.04	2.15 ± 0.06

## Data Availability

The dataset is publicly available at the University’s official repository.

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
