# Peer review of "Electrospray Deposition of Cellulose Nanofibers on Paper: Overcoming the Limitations of Conventional Coating"

_nanomaterials, 2021, doi:10.3390/nano12010079_

Round 1

Reviewer 1 Report

Dear editor,

Recommendation: Publish after major revisions.

Regarding the manuscript entitled "Electrospray deposition of cellulose nanofibers on paper: Over-coming the limitations of conventional coating".

In this paper, the authors reported a new strategy to enhance paper strength and air resistance by electrospraying cellulose nanofiber (CNF) dispersions. They compared the effects of electrospray deposition or bar coating of CNFs on the enhance of paper, and characterized the reinforced sheets by mechanical tests, Gurley porosity essays, scanning electron microscopy, Bendtsen roughness measurements, as well as routine determinations of the apparent density and the basis weight. On the whole, it is a meaningful work. However, the grammar, formats and analysis of the text need to be improved. The paper can be acceptable for publication in nanomaterials after major revision. And the following comments should be properly addressed before publishing this work.

  1. Introduction, the overall logic needs to be strengthened. For example, as a key enhanced composite, the merits and reinforcing mechanism of CNFs should be supplemented, and the advanced high-value applications of CNFs/CNCs/nanocellulose, such as in the fields of flexible green electronics and wearable human-machine interaction equipment, should also be mentioned to demonstrate the great potential of nanocellulose. So the following references are recommended to be mentioned in the introduction part, such as Advanced Materials, 2020, 2002264. doi/10.1002/adma.202002264; Nano Energy, 2021, 87, 106175. doi.org/10.1016/j.nanoen.2021.106175;
  2. Figure 1-6, the format of legend is recommended to keep consistent.
  3. “wt%.” and “wt.%”, the format should keep consistent.
  4. 3.1. Bulk addition of cellulose micro- and nanofibers and 3.2. Surface treatments with oxidized cellulose nanofibers, the stress-strain curves and the data comparison of mechanical performances should be supplied.
  5. 3. Results, the enhancement mechanism should be provided and can be characterized by means such as Fourier Transform Infrared Spectrometer.
  6. 3.2.1. Limitations of bar coating in the enhancement of paper properties, the spelling mistake of “umpredictable” should be corrected.
  7. References, there are many mistakes, such as lack of page numbers, no bold of year, etc., please check and calibrate uniformly.

Author Response

Dear Editor and Reviewer #1:

Here follows a detailed response to the first reviewer's comments to the manuscript nanomaterials-1482587. All changes pertaining to text editing have been marked in the revised version of the manuscript, using the “Track Changes” function as indicated in Ms. Wang’s e-mail. Please note that Figures have been modified as well.  

Recommendation: Publish after major revisions.

Regarding the manuscript entitled "Electrospray deposition of cellulose nanofibers on paper: Over-coming the limitations of conventional coating".

In this paper, the authors reported a new strategy to enhance paper strength and air resistance by electrospraying cellulose nanofiber (CNF) dispersions. They compared the effects of electrospray deposition or bar coating of CNFs on the enhance of paper, and characterized the reinforced sheets by mechanical tests, Gurley porosity essays, scanning electron microscopy, Bendtsen roughness measurements, as well as routine determinations of the apparent density and the basis weight. On the whole, it is a meaningful work. However, the grammar, formats and analysis of the text need to be improved. The paper can be acceptable for publication in nanomaterials after major revision. And the following comments should be properly addressed before publishing this work.

We are thankful for this opportunity to improve the quality of the manuscript. Regarding grammar and structure, please note that changes have been made all along the text. Some examples: “low value-added commodity” (much more common) instead of “low added-value commodity”, “by means of a Bendtsen tester” instead of “by bendtsen tester”, “hydrochloric acid” instead of “chlorhydric acid”.

Introduction, the overall logic needs to be strengthened. For example, as a key enhanced composite, the merits and reinforcing mechanism of CNFs should be supplemented, and the advanced high-value applications of CNFs/CNCs/nanocellulose, such as in the fields of flexible green electronics and wearable human-machine interaction equipment, should also be mentioned to demonstrate the great potential of nanocellulose. So the following references are recommended to be mentioned in the introduction part, such as Advanced Materials, 2020, 2002264. doi/10.1002/adma.202002264; Nano Energy, 2021, 87, 106175. doi.org/10.1016/j.nanoen.2021.106175;

The introduction has been enriched with comments in this regard, besides adding those references.

“The possibilities seem endless, from reinforcement of structural materials [6] to providing all kinds of electronic devices, such as transistors and nanogenerators [7,8], with flexibility and transparency.”

Figure 1-6, the format of legend is recommended to keep consistent.

Figures have been modified accordingly so that each variable can be easily identified. We understand that this lack of consistency particularly affected Figure 3, which has been reformatted.

“wt%.” and “wt.%”, the format should keep consistent.

In the revised version, we opt for “wt.%” consistently.

3.1. Bulk addition of cellulose micro- and nanofibers and 3.2. Surface treatments with oxidized cellulose nanofibers, the stress-strain curves and the data comparison of mechanical performances should be supplied.

We decided to include elasticity (Young’s modulus) just as a supplementary parameter, not highlighting performance in terms of stress-strain, because breaking length (or tensile strength / tensile index) is a much more common indicator of paper strength in the related literature. Monitoring the strain of paper is no easy task.

In other words, we took advantage of the fact that the universal tester reports several parameters from the same sample; otherwise, we would not have performed a specific test for the elasticity modulus. In works with structural materials such as cellulose-plastic composites, we opted for a more accurate and specific determination by means of a MFA 2 extensometer, and then we sometimes displayed the stress-strain curve. See e.g. doi:10.3390/polym12061308; doi:10.3390/polym10070699; doi:10.3390/polym13223947

The discussion on the mechanical performance has been enhanced, including comparison with other works. E.g.: Nonetheless, it should be noted that this enhancement of paper strength is in the range of what could be expected from refining to more than 1,500 PFI revolutions [39].

3. Results, the enhancement mechanism should be provided and can be characterized by means such as Fourier Transform Infrared Spectrometer.

In-depth characterization of CMNFs and oxidized CNFs has already been published in other works of ours, e.g.:

Sanchez-Salvador et. al. Enhanced Morphological Characterization of Cellulose Nano/Microfibers through Image Skeleton Analysis. Nanomaterials 2021, 11, 2077, doi:10.3390/nano11082077.

Serra-Parareda et al. Correlation between Rheological Measurements and Morphological Features of Lignocellulosic Micro/Nanofibers from Different Softwood Sources. Int. J. Biol. Macromol. 2021, 187, 789–799, doi:https://doi.org/10.1016/j.ijbiomac.2021.07.195.

Serra et al. Reducing the Amount of Catalyst in TEMPO-Oxidized Cellulose Nanofibers: Effect on Properties and Cost. Polymers (Basel). 2017, 9, 557, doi:10.3390/polym9110557.

In the revised version, the discussion has been enriched, mentioning, among other things, intermolecular interactions (mainly hydrogen bonds) as a reason for strengthening.

Revised version: Both refining and CMNF addition increase the specific surface area and thus the density of hydrogen bonding interactions, the main factor for paper strength.

3.2.1. Limitations of bar coating in the enhancement of paper properties, the spelling mistake of “umpredictable” should be corrected.

Corrected.

References, there are many mistakes, such as lack of page numbers, no bold of year, etc., please check and calibrate uniformly.

Consequences of trusting the reference management software. Somehow, RIS files were imported without page numbers and classified as items other than Journal Article. References have been corrected in the revised version.

Reviewer 2 Report

I have truly appreciated the approach, the way of presenting the experimental results and the discussion methodology, that properly and critically commented the performance of the paper sheet by comparing the standard bar layering and the novel electrospraying.

In my opinion, both scientific content and writing form can be cosnidered excellent, so I would suggest to consider publication of the manuscipt in its present form.

Author Response

Dear Editor and Reviewer #2,

Here follows the response to the reviewer's comments.

I have truly appreciated the approach, the way of presenting the experimental results and the discussion methodology, that properly and critically commented the performance of the paper sheet by comparing the standard bar layering and the novel electrospraying.

In my opinion, both scientific content and writing form can be cosnidered excellent, so I would suggest to consider publication of the manuscipt in its present form.

We are thankful to the reviewer, hoping that the revised version is even better, both in terms of content and form.

Reviewer 3 Report

Dear authors!

While the results of your research are definitely of interest for the scientifc community, I still have some major reservations regarding the following aspects:

  1. Apparently unrefined pulp was used in this study, which was never explicitly stated anywhere in the paper. Actually I do not understand this, as in industrial papermaking chemical pulps are always refined. Thus, especially the mechanical, but as well as all other properties of your reference hand sheets are at a very low level, as the pulp fibres are neither flexibilised nor fibrillated. As you are certainly well aware, similar (or even better) properties as the ones you obtained by bulk addition of CMNFs could easily also have been obtained by simply refining the pulp due to flexibilsation and fibrillation of the fibres.
    You do not mention this anywhere in the paper, but on the other hand are over-enthusiastically talking about a more than 100% increase in stength and are also hinting in the introduction, that CMNFs could be the solution to improve the profitability of the paper industry as a whole. Therefore, I would strongly recommend to reconsider all statments made in your paper taking into account that your pulp was unrefined. Would you have conducted the same experiments using a refined pulp the improvments would certainly be not as significant.
  2. The methods you used with regards to sheet forming  are not described adequately, so based on your description it would be impossible to reproduce your results. Regarding sheet forming, please also clarify how you made sure that the amount of CMNFs added were actually retained in the sheet. As you only state that 76 gsm handsheets were produced and nowhere in the paper show any basis weights of the produced handsheets, it is difficult to judge whether you really managed to retain the CMNFs (There was an interesting paper by Giner Tovar et al. published in Bioresources a few years ago, who discussed the issue of how you can make sure that the fine material you add is actually retained in the sheets).
  3. What is even more problematic is that your method of sheet drying was not specified at all anywhere in your paper. I would assume you used the Rapid Köthen dryers to dry the sheets with bulk addition of CMNFs, but I have no idea how you dried the bar-coated sheets or the electrosprayed sheets, which - as you also discussed for the bar-coated samples - definitely will influence the results. Acutally I am not sure whether the results you obtained in bar coating and electrospraying regarding the increase of roughness or the loss in strenght in bar coating (as you mentioned) are not caused by the effects of rewetting. So you definitely have to take this influence into account - also for electrospraying.
  4. Regarding the air resistance I actually would consider the achievement of a Gurley value of a bit above 300s with electrospraying quite poor. Also the improvement of around 100 Gurley s compared to bar coating appears quite disappoiniting to me  and I would be interested to understand, why the value is so low given the excellent coverage demonstrated by your SEM images. At this Gurley level you are definitely far away from having produced a paper sample that could be measured regarding OTR, whereas other authors (e.g. the groups of J. Bras or W. Bathchelor) have managed to achieve quite good OTR values at similar addition levels of CMNF to the paper surface.
  5. You are actually hinting in your paper that electrospraying could be a viable solution to improve the quality of industrially produced paper and board, but you are totally neglecting the fact, that electrospraying or electrospinning have by far not yet reached a development stage so that they could be applied industrially in the paper industry (to my knowledge we are talking presently of several hundreds of grams per hour to be produced in electrospraying compared to several tens of tons produced in industrial papermaking). While electrospraying to my opinion therefore is not a technology to be considered for application in the paper industry, other direct coating methods such as curtain or spray coating could be more viable.

So overall I would ask you to reconsider the statements made in your paper taking into account the remarks above.

I will not give any remarks regarding the correction of details of your submitted paper as I would expect you to significantly revise your paper. Detailed remarks will follow for the revised submission.

Author Response

Dear Editor and Reviewer #3:

Here follows a detailed response to the third reviewer’s comments to the manuscript nanomaterials-1482587. All changes pertaining to text editing have been marked in red in the revised version of the manuscript, using the “Track Changes” function as indicated in Ms. Wang’s e-mail. Undoubtedly, this constructive criticism has prompted us to improve the quality of the manuscript and to clarify some important points that remained ambiguous, unspecified or misleading (technical viability, basis weight data, experimental details), in order to ensure its scientific soundness.

Given the length of this response letter, we believe the DOC file (attached herewith) will be easier to follow than this format. Just in case (different people, different ideas), the version for HTML is provided too. Here it goes.

Dear authors!

While the results of your research are definitely of interest for the scientifc community, I still have some major reservations regarding the following aspects:

  1. Apparently unrefined pulp was used in this study, which was never explicitly stated anywhere in the paper. Actually I do not understand this, as in industrial papermaking chemical pulps are always refined. Thus, especially the mechanical, but as well as all other properties of your reference hand sheets are at a very low level, as the pulp fibres are neither flexibilised nor fibrillated. As you are certainly well aware, similar (or even better) properties as the ones you obtained by bulk addition of CMNFs could easily also have been obtained by simply refining the pulp due to flexibilsation and fibrillation of the fibres.

You do not mention this anywhere in the paper, but on the other hand are over-enthusiastically talking about a more than 100% increase in stength and are also hinting in the introduction, that CMNFs could be the solution to improve the profitability of the paper industry as a whole. Therefore, I would strongly recommend to reconsider all statments made in your paper taking into account that your pulp was unrefined. Would you have conducted the same experiments using a refined pulp the improvments would certainly be not as significant.

As the reviewer rightfully understands, we are aware of the fact that chemical pulps need to be refined to attain acceptable strength for sheet forming. But this has some continuity with previous works: please see 10.1016/j.carbpol.2015.12.004, where one of the main points was precisely the comparison between refining and CNF as strength agent: “The increase in Tensile Index was equivalent to 1600 [PFI revolutions].”

Indeed, at least with the current technology, nanofibrillation requires a higher energy input than refining, so in practice CMNFs can complement, not replace, refining. In this direction, we could also note that our isotropic sheets (like in most papermaking articles) cannot equal the tensile strength, measured in the machine direction, of anisotropic sheets with a similar pulp. Or if the pulp was never-dried. So the point was shedding some light on the use of CMNFs/CNFs as reinforcement without hoping that a breaking length of 4.6 km, by itself, would impress papermakers.

We believe, however, that at the very least these results are really valuable in relative terms, i.e., to what extent micro- and nanofibers can turn a weak paper sheet into one of acceptable strength.

In the revised version, we give up on any unjustified enthusiasm when discussing the results, insisting explicitly in the unrefined character of the bleached kraft pulp.

Revised version: Nonetheless, it should be noted that this enhancement of paper strength is in the range of what could be expected from refining to more than 1,500 PFI revolutions [39].

2.The methods you used with regards to sheet forming  are not described adequately, so based on your description it would be impossible to reproduce your results. Regarding sheet forming, please also clarify how you made sure that the amount of CMNFs added were actually retained in the sheet. As you only state that 76 gsm handsheets were produced and nowhere in the paper show any basis weights of the produced handsheets, it is difficult to judge whether you really managed to retain the CMNFs (There was an interesting paper by Giner Tovar et al. published in Bioresources a few years ago, who discussed the issue of how you can make sure that the fine material you add is actually retained in the sheets).

As the corresponding author, it was my arbitrary decision to remove the basis weights from Table 1 right before submission, fearing that the little differences are not significant. They have been incorporated in the revised version, though. As in any other paper engineering lab, all handsheets produced in our laboratory are weighted after drying and have their thickness measured.

We understand that the referred Giner Tovar et al.’s paper is doi:10.15376/biores.10.4.7242-7251. Recirculating the white water until a “steady state” is reached is also of interest for CMNFs and we will consider this in future works.

Retention of CMNFs is an issue, of course, and the main reason why we use a dual retention aid (cationic starch + colloidal silica). This has been treated in more detail in previous works of the group, e.g. 10.1007/s10570-014-0473-2: “The application of these retention agents (…) becomes necessary in order to avoid the loss of CNF during the dewatering process since filters at the bottom of the stock container in the Rapid-Köthen equipment are not able to retain nanometric material.” Actually, the use of this dual system with a pulp/CMNF blend and a standard former was comparable to the retention of nanofibers on a nitrocellulose membrane (<1 μm).

In the revised version, besides displaying the basis weights, the importance of CMNF retention is briefly discussed.

Revised version: Nonetheless, their total retention in the sheet cannot be granted since, as in the case of pulp fines, CMNFs are too small to be mechanically retained [10,36]. The insignificant differences in basis weight drive us to infer that the retention aid system succeeded in binding pulp fibers and most micro- and nanofibers together.

  1. What is even more problematic is that your method of sheet drying was not specified at all anywhere in your paper. I would assume you used the Rapid Köthen dryers to dry the sheets with bulk addition of CMNFs, but I have no idea how you dried the bar-coated sheets or the electrosprayed sheets, which - as you also discussed for the bar-coated samples - definitely will influence the results. Acutally I am not sure whether the results you obtained in bar coating and electrospraying regarding the increase of roughness or the loss in strenght in bar coating (as you mentioned) are not caused by the effects of rewetting. So you definitely have to take this influence into account - also for electrospraying.

Indeed, Rapid Köthen dryers were used after bulk addition. As for coated sheets, they were first air-dried by means of a thermo ventilator, while attached to a steel surface, and then briefly heated-pressed (1 min) in a Rapid Köthen dryer before additional coating processes. This has been made explicit in the revised version.

In the Discussion section, the loss of strength is being attributed to rewetting and redrying. We understand this leads to fiber collapse / hornification.

  1. Regarding the air resistance I actually would consider the achievement of a Gurley value of a bit above 300s with electrospraying quite poor. Also the improvement of around 100 Gurley s compared to bar coating appears quite disappoiniting to me and I would be interested to understand, why the value is so low given the excellent coverage demonstrated by your SEM images. At this Gurley level you are definitely far away from having produced a paper sample that could be measured regarding OTR, whereas other authors (e.g. the groups of J. Bras or W. Bathchelor) have managed to achieve quite good OTR values at similar addition levels of CMNF to the paper surface.

Certainly, for a truly sealed paper the time spent by 100 mL of air to pass through should be at least one order of magnitude higher. Nonetheless, despite the apparently homogeneous (but thin) coverage, the coat weight was only ~3 g/m2 and the paper sheet itself offers little resistance. In fact, the reference paper is very permeable to begin with (<2 s). Even in the case that there was some systematic error (e.g., much less than 100 mL of air effectively passing), i) it equally affected all samples, and ii) after all, these results are in the same range as the ones reported by Mousavi et al. (2017) with refiner CNFs (rod coating) if the coat weight was 3.3 g/m2.

Revised version, Results: still in the medium range but taking over 200 times longer than control sheets.

Revised version, Discussion: The resulting values are in the same range or one order the magnitude lower than those of Mousavi et al. [41], who applied CNFs and carboxymethyl cellulose on paperboard by rod coating, but their starting material offered a much higher air resistance (60 Gurley s). The resulting values are in the same range as those of Mousavi et al. [41] for rod coating of refiner CNFs with a similar weight gain, but one order of magnitude lower than when they added carboxymethyl cellulose and/or obtained higher coat weights.

  1. You are actually hinting in your paper that electrospraying could be a viable solution to improve the quality of industrially produced paper and board, but you are totally neglecting the fact, that electrospraying or electrospinning have by far not yet reached a development stage so that they could be applied industrially in the paper industry (to my knowledge we are talking presently of several hundreds of grams per hour to be produced in electrospraying compared to several tens of tons produced in industrial papermaking). While electrospraying to my opinion therefore is not a technology to be considered for application in the paper industry, other direct coating methods such as curtain or spray coating could be more viable.

So overall I would ask you to reconsider the statements made in your paper taking into account the remarks above.

The reviewer is right, but we believe that electrospray deposition, when it comes to feasibility, lies between the coating procedures that are well-known to the industry (bar coaters, size press, curtain coating, doctor blades, spray nozzles, etc.) and the ways which are least likely to be scaled up to a paper mill (RF magnetron sputtering, chemical vapor deposition, etc.). Furthermore, the results of electrospray deposition could be somehow extrapolated to encourage other kinds of CNF spraying.

Admittedly, we were intentionally ambiguous on this matter since, while papermakers do not use this technique, patents protecting electrospray coating have been being filed since the one assigned to Minnesota Mining And Manufacturing Company in 1987. Then, even if devoid of feasibility for the paper industry, electrospray deposition is not devoid of industrial interest.

In the revised version, however, we are explicit on the lack of industrial feasibility, at least in the near future.

Revised version, Introduction: and partially because electrospray deposition still seems unthinkable in a paper mill

Revised version, Conclusions: While electrospray deposition is currently not feasible for a paper machine, an adaptation of the concept to the existing technology (e.g., spray nozzles) could be encouraged.

I will not give any remarks regarding the correction of details of your submitted paper as I would expect you to significantly revise your paper. Detailed remarks will follow for the revised submission.

We are looking forward to it, so as to keep improving the quality of the manuscript!

Round 2

Reviewer 1 Report

The authors have addressed all the questions properly, so the paper can be published.

Author Response

We thank the reviewer for the recommendation of acceptance.

Reviewer 3 Report

Dear authors!

Thank you for considering my main remarks.

As stated in my previous review, please find below some further necessary corrections:

General:

A) Please make sure to be clear whether the given percentages are weigth% or volume%.

B) Please state somewhere whether the deviations given in the tables are standard deviations or confidence intervals

C) Please add error bars in all figures!!!!

Detailed corrections and remarks (numbers refer to line number):

14: " ....bulk addition to the unrefined pulp was enough to attain..."

19: "...allowing enhancements... " (delete for notorious")

31: References 4+5 are not really on CMNF production - there would be better ones

35: "annual growth rate of up to 21.3%"

36-38: Has to be changed, as the referenced paper [10] does not say this. Rephrase to :" Not surprisingly, among all the industries that could potentially benefit from the use of this renewable nanomaterial, paper companies play an important role"

39: As you have quite a high number of self-citations in your paper, delete citation 11 and use one of the many other publications on this topic

42: "....seems to be a promising approach."

45: "...amounted to only 22%...."

46: "...to increase this benefit": Which benefit are you referring to - be more specific!

54: "...has severe limitations"

57: "...of e.g. polyethylene..."

62-64: Your reference to pulping is unclear - what exactly do you want to say with this sentence?

66-67: "As advocated in previous work, TEMPO...."

70:  "...work, we suggest..."

91-92: "...an unrefined bleached hardwood kraft pulp (BHKP)" (that is how this pulp is commonly referred to"

93-94: length and fibre widths was measured how?

104: specify the exact type of cationic starch and colloidal silica !!!

112: BHKP

Section 2.3 general:

It is important to state wheter cationic starch and silica were also added to the reference sheets containing no nanofibers, as CS potentially also has a strengthening effect as such.

135: "were added to the bleached hardwood kraft pulp suspension..." - Please also state the conistency of the pulp suspension!

142: weight

144-145: Please state air temperature and duration of air drying. Was it restrained drying or were the sheets able to shrink. At what dry content were they brought to the Rapid-Köthen dryers???? This is vital infomation in order to interpret your results!

146-147: How were the electrosprayed sheets dried (also state temperature and duration - this is important to understand the differences you later discuss in the results section!!!!)

159-161: "..resemble a viscous fluid ..." It is difficult to understand what you want to say with this statement. Also it is quite clear that you will not be able to distinguish nanofibres at the magnifications you used. Please try to rephrase this statement!

173-181: Change to: "While total retention of CMNFs in the sheet cannot be granted since CMNFs are too small to be mechanically retained [10,36],  the insignificant differences in basis weight allow the conclusion that the retention aid system succeeded in retaining most of the CMNFs. Regarding the surface, even without coating, Bendtsen roughness was slightly but consistently increased by the bulk incorporation of CMNFs. 

Regarding the surface, even without coating, Bendtsen roughness was slightly but consistently increased by the bulk incorporation of CMNFs. Figure 3 shows...."

Table 1: Please add deviations for roughness (as in Table 2)

196: "...in the latter case,..."

Caption Figure 4: 205 "at a magnification" 209: Please state magnifcation for corss-sectional fibre images. Also state, at which location these images have been taken (near surface of the sheets???)

221: "...i.e. a decrease in surface ..."

240-241: "Gurley air restistance reaches values over 300 s which is still in the medium range but significantly higher than the unrefined reference sheets". Again - do not overexaggerate when comparing to unrefined pulp!

242: "..., also worth mentioning.."

246: "...the performance of the unrefined reference pulp"

265: "..and thus (hydrogen) bonding..."

273-276: Please rephrase this statement, as it is unclear. What do you mean with fibres collapse one with another and I also do not think that the main reason is hornification. Consider, that be adding 1 g/m2 dryCMNF (0,45 wt.% solids) you are adding more than 200 g/m² of water, bringing down the dry content of your sheet to about 25% - therefore it is so important to know how your sheets were dried (restrained, free shrinkage etc).

280: I would argue that paper became rougher because so much water was added which led to some shrinkage and thus the roughness increase. In general adding something to the paper surface should decrease roughness - so reconsider this statement.

281: "Air permeability was decreased significantly, indicated by a significantly higher value of Gurley air resistance compared to the reference case (300 Gurley s vs. 1.5 Gurley s).

293-295: Unclear sentence - please rephrase (esp. redrying negatively affected filtering and bar coating???)

307: Delete "Fortunately,"

308-310: "The applied nanofiber layer not only slowed down air flow, attaining a Gurley air resistance of up to 300 s, but also led to a significant increase in the breaking length especially when compared to the unrefined reference handsheets with no bulk addition of CMNFs." Please do not always write 200x higher and triplicate - we are scientists and not marketing people!

Best regards

Author Response

Dear Editor and Reviewer #3:

Considering the length of the detailed point-by-point response (6 pages), this is to be found as a PDF file —attached herewith. All of the third reviewer’s comments to the manuscript nanomaterials-1482587_R1, following requisition of minor revisions, are addressed and have prompted changes in the manuscript. All changes have been marked in the revised version of the manuscript, using the “Track Changes” function as indicated in Nanomaterials' e-mail. Once again, this constructive criticism has prompted us to keep improving the soundness of the manuscript.
